# Circadian Rhythm Disruption as a Contributor to Racial Disparities in Prostate Cancer

**DOI:** 10.3390/cancers14205116

**Published:** 2022-10-19

**Authors:** Sonali S. Dasari, Maddison Archer, Nihal E. Mohamed, Ashutosh K. Tewari, Mariana G. Figueiro, Natasha Kyprianou

**Affiliations:** 1Department of Urology, Icahn School of Medicine at Mount Sinai, New York, NY 10029, USA; 2Tisch Cancer Institute, Mount Sinai Health, New York, NY 10029, USA; 3Department of Oncological Sciences, Icahn School of Medicine at Mount Sinai, New York, NY 10029, USA; 4Light and Health Research Center, Department of Population Health Science and Policy, Icahn School of Medicine at Mount Sinai, New York, NY 10029, USA

**Keywords:** racial disparities, prostate cancer, circadian genes, night shift work, artificial light at night, jet lag, obesity, stress, melatonin, treatment resistance

## Abstract

**Simple Summary:**

African American (AA) men have 2.4 times higher mortality rate due to prostate cancer than White men in the United States. Evidence implicates circadian rhythm disruption (CRD) as a potential driver of prostate cancer risk and progression. AA men are particularly vulnerable to CRDs due to greater exposure to night shift work, artificial light at night, noise pollution, racial discrimination, and socioeconomic disadvantages. In this review, we discuss the growing contribution of CRDs to the racial disparities associated with the incidence, aggressiveness, and progression of prostate cancer and highlight the unmet clinical need of integrating circadian-related therapies to enhance current prostate cancer treatment modalities.

**Abstract:**

In the United States, African American (AA) men have a 2.4 times higher mortality rate due to prostate cancer than White men. The multifactorial causes of the racial disparities in prostate cancer involve various social determinants of health, socioeconomic status, and access to healthcare. However, emerging evidence also suggests that circadian rhythm disruption (CRD) contributes to prostate cancer, and AA men may be more susceptible to developing CRDs. Circadian rhythms play a significant role in metabolism, hormone secretion, and sleep/wake cycles. Disruption in these circadian rhythms can be caused by airplane travel/jetlag, night shift work, exposure to light, and neighborhood noise levels, which can contribute to sleep disorders and chronic conditions such as obesity, diabetes, cardiovascular disease, and depression. The drivers of the racial disparities in CRD include night shift work, racial discrimination, elevated stress, and residing in poor neighborhoods characterized by high noise pollution. Given the increased vulnerability of AA men to CRDs, and the role that CRDs play in prostate cancer, elucidating the clock-related prostate cancer pathways and their behavior and environmental covariates may be critical to better understanding and reducing the racial disparities in prostate cancer.

## 1. Introduction

Prostate cancer (PCa) is the second leading cause of cancer-related mortality for men in the United States [1,2]. Significant racial disparities exist at all stages of PCa treatment, including diagnosis, management, and follow-up care [3]. African American (AA) men are 1.6 times as likely to be diagnosed with PCa and at 2.4 times higher risk to die of the disease compared to White men [4]. The racial disparities in PCa may be due to lower rates of health literacy, comorbidities such as diabetes and obesity, and behavioral factors such as smoking; in addition, racial bias and barriers to healthcare access impact PCa outcomes of AA men [3]. While the current risk factors for PCa are age, race, and family history, increased epidemiological evidence points to the role of circadian rhythm disorder (CRD) in cancer progression [5]. In addition to cancer metastasis, CRD is a risk factor for various other conditions, such as chronic sleep deprivation, obesity, metabolic syndrome, cardiovascular diseases, and psychiatric diseases [6]. CRD is characterized by the lack of synchrony between the endogenous master circadian clock and the external light–dark cycles [7]. 

AA men and women may be at heightened vulnerability for developing CRD due to a greater prevalence of night shift work, environmental factors (e.g., living in neighborhoods characterized by high noise pollution, exposure to low daytime light levels or too much nighttime light), and chronic conditions (e.g., diabetes, obesity, long-term stress, cardiovascular disease) [8,9,10,11,12,13]. The underdiagnosis of sleep disorders, including obstructive sleep apnea (OSA), within predominately AA communities contributes to abnormal sleep architecture, which puts OSA patients at high risk for CRD. Further research is required to understand the exact relationship between CRD on OSA [14,15,16,17]. Epidemiological evidence so far points to the connections between PCa and CRD. Meta-analyses on the health outcomes of airplane pilots [18,19] and studies on female night shift nurses [20] have demonstrated the contribution of CRD (due to jetlag or night shift work) to both prostate and breast cancer [21]. Sleep disruption and light-induced melatonin suppression, both related outcomes of CRD, are associated with an increased risk for advanced PCa [22,23]. Lastly, the consequences of CRD—including circadian gene polymorphisms and conditions such as diabetes, obesity, and depression—contribute to an increased PCa risk [24,25,26,27,28].

Taken together, the unique impact CRD has on the AA population and its role in PCa, elucidating the biological mechanisms through which CRD contributes to PCa, may be critical to mitigating the racial disparities in PCa outcomes. Compelling evidence suggests that CRD may contribute to PCa progression through (a) circadian-gene variants [29,30,31] (b) stress and obesity-related biological pathways [32,33,34], and (c) melatonin inhibition [35]. As a result of CRDs, circadian clock genes no longer function as tumor suppressors, contributing to worse PCa outcomes. The complex interplay between stress, obesity, and circadian disruption may have detrimental effects on the tumor microenvironment and could enhance the stress-related PCa growth pathway, otherwise known as the glucocorticoid-mediated androgen receptor signaling pathway. Considering these findings, targeting the melatonin pathway and the glucocorticoid receptor, both of which are implicated in CRD, may provide new opportunities to impair PCa growth and overcome therapeutic resistance, respectively. 

### 1.1. Regulation of the Circadian Clock System 

Driven by the 24 h rotation of the planet, almost all organisms have adapted on earth by developing an internal biological clock system known as the circadian system, which rhythmically synchronizes sleep, metabolic, and dietary behavior to light/dark cycles [36]. In any given tissue, around 10–20% of the genome is expressed in a circadian manner [37]. Circadian rhythms play a significant role in the sleep/wake cycle, metabolic function, and gene expression [38]. Disruption of circadian rhythms has major consequences on the body’s ability to regulate metabolic homeostasis [39]. 

The central circadian clock, located in the suprachiasmatic nuclei (SCN) of the anterior hypothalamus, regulates the timing of activities of the peripheral clocks [40]. Light–dark patterns synchronize the SCN with the external environment, assuring that the body does the “right thing at the right time” [41]. In diurnal species, the light phase is associated with an increase in body temperature, heart rate, and blood pressure. During the dark phase, melatonin production increases, body temperature declines, heart rate slows down, and blood pressure lowers [42]. Within individual cells, the circadian clock is self-regulated by transcriptional–translational feedback loops (TTFLs) [43]. TTFLs are comprised of a positive arm with a heterodimeric complex at its core that behaves as the activator of the system, promoting the transcription of one or more components of the negative arm, which, when translated, inhibits the activity of the positive arm. Transcription activator factors CLOCK and BMAL1 make up one arm of the feedback loop, and repressor proteins PER and CRY, made from *Per1/2/3* and *Cry1/2* genes, make up the other arm. Accessory TTFLs regulate the primary TTFL. The first accessory loop is made up of RORs and nuclear REV-ERB receptors, while the second accessory loop is composed of D-box-related genes and transcription factors, including albumin D-binding protein (DBP), thyrotroph embryonic factor (TEF), and hepatic leukemia factor (HLF) [7,44]. 

In humans, during the light phase (morning), transcription activators BMAL1 and CLOCK form heterodimers, bind to the E-box (5′-CACGTG-3′), and signal the transcription of target genes *Period (Per 1/2/3)* and *Cryptochrome*
*(Cry 1/2)* [45]. These target genes encode transcriptional repressors, PER and CRY proteins. During the light phase, PER and CRY transcription is high, and PER and CRY proteins accumulate in the cytoplasm. During the dark phase (evening), PER and CRY proteins dimerize to form PER–CRY complexes, with subsequent nuclear translocation and inactivation of the CLOCK/BMAL1-mediated transcription, reprising their own transcription and closing the loop. As the dark phase progresses, PER and CRY complexes are gradually phosphorylated by casein kinase I (CkIδ and CkIε) and 5′ AMP-activated protein kinase (AMPK), and subsequently realize their degradation through the proteasome pathway [46]. Degradation of the PER and CRY repressor proteins allows for CLOCK-BMAL1 transcription to resume, thus initiating a new transcriptional cycle [47]. In addition to the primary feedback loop, accessory loops are formed from nuclear orphan receptors, retinoid-related orphan receptors (RORs), and REV-ERBα/β, which target BMAL1 production through binding to ROR- binding elements (ROREs) [7]. REV-ERBα/β inhibits BMAL1 transcription upon binding, while ROR, acting as a positive regulator, initiates BMAL1 transcription [45] (Figure 1). 

While the SCN mainly relies on cues from light–dark cycles to entrain the biological clock, peripheral clocks, such as the ones in the reproductive, endocrine, and immune systems, receive timing signals from the SCN as well as from feeding patterns, temperature, and hormones [48,49,50]. The feed/fasting cycle dictates nutrient intake within specific periods during the day: the periodic phosphorylation of energy sensors such as AMP-activated protein kinase (AMPK) promotes ATP production in response to exercise/fasting and encourages the breakdown of fatty acids, glucose, and triglycerides after eating a meal. AMPK destabilizes CRY1 in peripheral proteins and interacts with SIRTUIN 1(SIRT1), which modulates transcription factors including PER2 [51]. In addition to nutrient-sensing molecules, ligand-activated transcription factors can have several effects on clock genes; REV-ERB regulates gluconeogenesis and the lipid metabolism while repressing *Bmal1* transcription, while transcription factor ROR, its competitive inhibitor, induces *Bmal1* expression upon binding. PPAR is activated by fatty acids and plays a role in lipid homeostasis through a positive feedback loop with BMAL1 protein [52]. Thus, peripheral clocks are involved in several important metabolic functions, including digestion, hormone secretion, lipid homeostasis, and the immune system response [53]. 

### 1.2. Epidemiological Evidence—The Link between CRDs and Prostate Cancer Risk 

Disruption of circadian rhythms is thought to be caused by environmental noise pollution [54,55], jet lag [56,57], night shift work [58], and artificial light at night (ALAN) [59]. CRD is associated with various health consequences, including premature death, metabolic syndrome, obesity, immune dysregulation, reproductive problems, stress, and depression [6]. Compelling evidence demonstrates that PCa is linked with both the causes and consequences of CRDs [60]. 

#### 1.2.1. Causes of CRD

Noise pollution: Nocturnal environmental noise, such as noise from transportation or industrial plants, can induce disturbances in sleep quality, metabolic and psychiatric changes, and alterations in sleep architecture [61]. As a consequence of noise exposure, the redistribution of time spent in different sleep stages—increasing wakefulness and stage one sleep, and decreasing slow-wave sleep and REM sleep—negatively impacts cognitive performance, mood, and energy restoration while increasing daytime sleepiness [62]. Epidemiological studies have found that exposure to traffic noise at night increases the risk for hypertension, heart disease, and stroke. Nocturnal noise contributes to increased risk for cardiovascular comorbidity through the greater secretion of endocrine hormones, including cortisol, noradrenaline, and adrenaline [61]. Nocturnal ambient noise exposure is associated with dysregulation of the central circadian clock, and may be involved in alternations in peripheral clock genes [63]. 

Jetlag: CRD may contribute to PCa through jetlag. Studies involving US pilots and astronauts have determined there is an increased risk of developing PCa, yet not PCa mortality [64,65,66]. Longer air hours, number of employment years, and radiation exposure positively correlate with increased PCa risk [19,67]. Male pilots are at least twice as likely to develop PCa than men in the general population, and several subgroups, including AA pilots and military pilots, were found to have an increased riskforf PCa [18,68]. 

Night shift work: In addition to jetlag, numerous studies have established the impact of night shift work on cancer risk. In 2017, the International Agency of Research on Cancer identified rotating shift work, in association with circadian disruption, as a probable human carcinogen, placing it in same risk category as ultraviolet radiation, benzo(a)pyrene, and acrylamide [69]. Co-exposures within night shift workplaces, including noise levels and light at night, may additionally be linked with CRD [70]. A longitudinal study found that female nurses who engaged in nightshift work for over thirty years displayed a 36% increase in relative risk of breast cancer [71]. The established evidence for the effect of night shift work on the prevalence of breast cancer has incited the investigation of night shift work in relation to PCa [72]. Several meta-analyses, including cohort-based studies in Japan, Canada, and Spain, found an association between night shift work and PCa [21,72,73,74,75]. In the Spanish cohort, workers with a longer duration of work hours were more likely to have tumors with a worse prognosis, while the Canadian population study determined that night-workers were at increased risk for developing PCa, regardless of work duration [74,75]. Additionally, a fixed vs. rotating night shift work has a differential effect on PCa risk, with rotating shift workers having a 20% higher risk for developing PCa than fixed schedule night shift workers [76]. Greater PCa risk has been reported in firefighters, health practitioners, and police, all of which typically require some degree of night shift work [77]. AA firefighters, machinery maintenance workers, and railroad workers are particularly more at risk for developing PCa [65]. Despite evidence implicating a positive correlation between night shift work and PCa, some reports found no such association [78,79]. These diverse findings may be related to the size of the cohort, differences in fixed schedules vs. rotating schedules, and duration of occupation. 

Artificial light at night: While the current literature on the effects of ALAN as an environmental risk factor are limited, there is evidence for the association between ALAN and PCa incidence [80,81]. The established literature on the link between ALAN exposure and risk of breast cancer has promoted further exploration of the effects on other hormone-dependent cancers, including PCa [80,82,83,84,85]. Several cross-geographic studies have found a significant positive association between population exposure to ALAN and the incidence of prostate, breast, colorectal, and lung cancers individually, after adjusting for population size, electricity consumption, air pollution, and total area of land covered by forest [86,87]. A case–control study in Spain found that night shift workers had a slightly higher prostate cancer risk compared to non-night shift workers; the risk increased with longer light exposure and was more pronounced for high-risk prostate tumors [75]. Exposure to ALAN affects melatonin levels, a potential mechanism linking shift work with increased PCa risk [88]. Lower melatonin serum levels have been associated with exposure to ALAN [89]. Thus, the lack of melatonin, a circadian hormone with potential anti-cancer effects, may be correlated with enhanced tumor development. A direct link between ALAN, melatonin suppression, and increased risk for cancer, however, has not been established, mainly due to a lack of measurements of ALAN using calibrated personal light-measuring devices.

#### 1.2.2. Consequences of CRD

Behavioral stress: Behavioral stress and psychosocial factors, which has a bidirectional relationship with CRD, have an impact on PCa outcomes [90]. PCa patients report the highest levels of stress and anxiety on average compared to other cancer patients. Findings from studies on patient anxiety, stress, and prostate specific antigen (PSA) levels are heterogenous [91,92,93]. Clinical studies have found that participants with high cortisol levels had a positive correlation with high prostate specific antigen (PSA) values, indicating high PCa risk [94]. Among a cohort of World Trade Center responders during the 9/11 terrorist attacks, re-experiencing a traumatic event was correlated with increased PCa incidence [95]. Greater perceived stress is associated with increased PCa-specific mortality, grieving and sleep loss, and a lack of adequate social support [96]. Patients utilizing high-effort coping (a coping mechanism used for race-based discrimination and mistreatment) had a slightly greater PCa risk relative to men who had decreased levels of high-effort coping, demonstrating how race-based discrimination may be a contributing factor to the stress-related PCa pathway [97]. Lastly, glucocorticoid (GR) signaling, which is overstimulated in chronic stress conditions, contributes to the progression of metastatic-castration-resistant prostate cancer (mCRPC), by promoting AR-target genes in the absence of androgens [98]. 

Obesity: Obesity is associated with multiple chronic problems as well as hormonal changes, which have an impact on the progression of PCa. Studies investigating the potential role of obesity in PCa risk, progression, and mortality yield heterogeneous results. Measures of obesity, such as body mass index (BMI) > 30, waist–hip ratio, and waist circumference (WC), are positively correlated with risk for advanced PCa and mortality due to PCa [99,100,101,102]. Obesity has a stronger association with PCa risk among AA men compared to White men [103]. Among a racially diverse cohort, AA men had the greatest obesity rates (BMI > 35), and obese AA men were more likely to have positive surgical margins, higher-grade tumors, and higher rates of biochemical failure after a radical prostatectomy [104]. A higher BMI has been associated with increased progression towards mCRPC, a 3-fold risk of developing metastases, and PC-specific mortality [105]. These findings provide new insights into the role of CRD and the consequences of CRD (e.g., obesity and stress) for PCa progression.

### 1.3. Racial Disparities in Circadian Health—Implications for Prostate Cancer 

In the United States, the AA population may be at increased risk for CRD due to occupational, environmental, and healthcare-access-related factors. The risk factors for CRD, such as night shift work [8], light at night [106], environmental noise pollution [62], and its consequences, including obesity, stress, and sleep deprivation, disproportionately affect AAs and may exacerbate existing racial health disparities [107] (Figure 2). A longitudinal study of White, Chinese American, Black, and Hispanic adults found that sleep irregularity was greater among Black participants compared to all other groups; greater sleep irregularity was also correlated with greater obesity, hypertension, and increased depression severity and perceived stress [108]. 

Residential segregation is a fundamental contributor to racial disparities in health outcomes and may contribute to fragmented sleep among racial/ethnic minorities. This form of discrimination was legitimized through “redlining”, a historical housing segregation policy that entitled banks to deny mortgages based on race and grant loans for Black and Brown communities only within hazardous (less desirable) parts of the city [109]. Segregated urban neighborhoods are more likely to have under-resourced schools, poor housing quality, a higher police presence, decreased accessibility to healthy food options, and greater stressful conditions, all of which may contribute to poor sleep conditions for ethnic minorities [11,12]. Historically redlined neighborhoods are closer in proximity to industrial plants, highways, and factories, and are characterized by overcrowding and thin walls, both of which contribute to greater levels of environmental pollution and noise levels [110]. One study found that neighborhoods with at least 75% Black residents had nocturnal noise levels four decibels higher compared to neighborhoods without any Black residents [13]. Marginalized groups, such as racial minorities, low income groups, and those with lower educational levels, experience the greatest noise exposure [10]. 

Circadian misalignment varies across racial groups and occupations. AA and Hispanic populations are disproportionately represented in night shift work; in fact, a study found that they are twice as likely to be working the night shift compared to their White counterparts [111,112]. The prevalence of comorbidities, such as obesity, cardiovascular disease, diabetes, and psychosocial stress, uniquely impacts AA communities [113]. Neighborhoods characterized as food deserts and food swamps, which refer to an urban area with limited access to healthy food and a high-density of junk food, are positively associated with obesity [9,114]. Additional experiences of racial discrimination and workplace harassment are associated with impaired sleep levels [115]. High crime rates and social isolation in segregated neighborhoods have a cumulative impact on health outcomes. Black populations in segregated neighborhoods are more likely to work night shifts, experiencing greater psychosocial stress compared to White populations [116]. 

The genetic variation in circadian periods may contribute to differences in the severity of consequences of CRD. On average, AAs have a shorter free-running circadian period, or *tau*, compared to European Americans, by 0.2 h. In response to circadian disruption, AAs are found to have shorter *tau* and thereby shorter phase delays, which makes it more difficult to adapt to night shift work and contributes to longer jetlag, on average, compared to European Americans, who were found to have larger phase delays [117]. Given that AAs have smaller phase delays and are overrepresented in shift work, their potential exposure to shift work and its negative health consequences is greater [118].

The underdiagnosis of sleep disorders in AA communities contributes to poor sleep quality and may lead to increased risk for CRD. AA adults are more likely to experience less deep (slow wave) sleep, twice as likely to be short sleepers, and take longer to fall asleep compared to their White counterparts [119]. Access to health insurance in relation to sleep outcomes is not well studied. However, some studies have found that lower access to private healthcare is associated with greater gasping during sleep, and those with more recent healthcare visits reported less daytime fatigue and sleep disturbance [120]. The shortage of primary care physicians near low-income neighborhoods may play a role in the underdiagnosis of sleep apnea within Black populations [121]. Obstructive sleep apnea is associated with sleep fragmentation due to repeated arousal throughout the night, which may contribute to its long-term consequences including daytime sleepiness, cognitive impairment, and cardiovascular problems [122]. In poor urban neighborhoods, increases in reported sleep apnea have been associated with increases in environmental pollution levels [123]. Changes in inflammatory markers that normally exhibit a circadian rhythmicity have been observed in OSA patients, and further research is necessary to elucidate the exact mechanism between sleep apnea and circadian rhythm disruption [16]. The combined effects of structural, environmental, and psychosocial conditions contribute to poor sleep architecture and greater risk for developing CRDs for marginalized groups and may disproportionately impact the AA population. 

### 1.4. The Consequences of CRD for Prostate Cancer Risk and Progression 

Given the increased vulnerability of AA men to developing CRD and the epidemiological impact of CRDs on (both PCa causes and consequences), understanding the CRD-mediated cancer pathways may facilitate reduction in the racial disparities in PCa mortality and incidence. Compelling evidence suggests that CRDs contribute to PCa progression through (a) circadian gene variants (b) stress and obesity-related biological pathways, and (c) melatonin inhibition (Table 1). 

Circadian Gene Variants. At the cellular and molecular levels, dysregulation of circadian clock machinery, which functions as a tumor suppressor and regulates tumor growth, has been shown to play a role in cancer development [138]. There is significant crosstalk between clock genes and cell cycle, apoptosis, DNA repair, and senescence [139]. On a systematic level, dysregulation of the circadian system could disrupt the immunosuppressive function of clock-mediated pathways and tumor inhibition. At the molecular level, the SCN regulates the expression of clock-controlled genes in cells of peripheral tissues, including c-Myc, Mdm2, as well as tumor suppressor gene Trp53. Intracellular signaling pathways directly involved in cell proliferation, such as the B-catenin WNT signaling pathway, are correlated with circadian regulation [138]. 

Epidemiological evidence suggests that the circadian pathway is significantly associated with advanced PCa among night shift workers, particularly those who worked for at least 20 years and had long shifts; at the gene level, the night shift workers exhibited altered *BMAL1, NPAS2*, and *RORA*, which is associated with aggressive PCa, yet there were not significant associations found at the SNP-level [24]. Men with *CRY2* variant C allele were almost 1.7 times as likely to develop PCa in comparison to men with the *CRY2* normal allele [140]. Night shift workers had significantly increased nocturnal levels of PER2 compared to day shift workers, perhaps demonstrating the potential of PER2 as a circadian biomarker to reflect changes in the peripheral circadian rhythm associated with PCa [141]. Due to these changes, shift workers have increased risk for metabolic syndrome, cognitive disorders, sleep disorders, and lower melatonin levels [142,143].

Circadian gene expression has a direct correlation with PCa risk. *PER1, PER2*, and *CLOCK* levels are downregulated, whereas *BMAL1* levels are unregulated in PCa tissue compared to normal tissue. Overexpression of *PER1* and *PER2* in PCa cell growth suppression occurs via apoptosis, suggesting that Per 1-2 exerts anti-tumor effects [124]. Population-based genetic association studies which genotyped single-nucleotide polymorphisms (SNP) in known circadian-related genes found that alterations in at least one SNP in each of the core circadian genes was correlated with PCa risk. However, there are variations in the strength of association for fatal PCa compared to risk for PCa and variations in SNPs among different cohorts [124]. Among SNPs from nine circadian-related genes, a case–control study of Caucasian men with PCa revealed that levels of *Per1-3, CSNK1E, Cry1-2, Arntl, Clock*, and *NPAS2* SNPs were significantly associated with risk for aggressive PCa. These findings suggest the potential value of identifying genetic variants of circadian genes in order to quantify risk for PCa [29]. However, the genetic variations in circadian rhythm and melatonin pathways have only largely been investigated in non-racially diverse populations, making it difficult to generalize these results to the entire population. 

PCa is dependent on the androgen receptor (AR), and first-line treatments for patients involve androgen depletion and AR-targeted therapies. However, patients eventually develop resistance, and an AR-dependent form of PCa, called castration-resistant prostate cancer (CRPC) emerges for which there is no cure. The development of novel strategies to address the challenges of CRPC alongside AR-targeted therapy are required. Recent genome-wide studies reveal that *CRY1*, which regulates DNA repairing and cell proliferation, has relevance to advanced disease as an AR-regulated and pro-tumorigenic factor that promotes DNA repair and the survival of cancer cells via AR binding to the *CRY1* locus [30]. Given that CRY1 levels are androgen-sensitive and predict poor PCa outcomes, CRY1 may serve as a therapeutic target for late-stage disease [30]. PER3 has been identified as a negative regulator of PCa stem cells (PCSCs) through the activation of the WNT/Beta-Catenin signaling pathway. PER3 is downregulated in human Pca clinical samples. Overexpression of PER3 in PCa-resistant cells inhibits cell proliferation and stimulates BMAL1 expression, leading to the inactivation of the WNT/Beta-Catenin pathway [31]. Additionally, PER1 is downregulated in clinical PCa samples compared to normal tissues. The binding of PER1 to AR was shown to decrease androgen-sensitive genes in the presence of androgens, and the overexpression of PER1 inhibited proliferation and increased apoptosis. These results suggest an impact of PER1 circadian disruption on AR-mediated PCa development and the benefits of chronotherapy for optimizing current Pca treatments [126]. An SNP of the *NPAS2* gene, rs6542993, was associated with greater risk for disease progression in patients with localized PCa, indicating the value of *NPAS2* SNP as a potential biomarker for PCa progression [144]. Despite the positive correlation between circadian gene variants and PCa risk, some studies have found no correlation between circadian clock gene SNPs and incidence of lethal PCa [145,146]. In response to neoadjuvant enzalutamide treatment (a second-generation antiandrogen treatment), BMAL1 (also known as *Arntl*), a circadian rhythm core component, was found to be the second-most-enriched transcription factor after FOXA1. The dependence on BMAL1 as a transcriptional regulator of cellular proliferation may describe the mechanism through which tumor cells evade AR blockades [147]. Given the potential of BMAL1 as a novel therapeutic target, future research is required to develop strategies to inhibit this process to further increase the efficacy of anti-hormonal therapy. 

Stress-related Biological Pathways. There is significant crosstalk between the HPA axis and the central circadian clock system, which may have consequences for prostate cancer. The SCN communicates with the sympathetic nervous system to regulate the rhythmic secretion of glucocorticoid hormones from adrenal glands. Glucocorticoids act on peripheral clocks located in the adrenal glands and stimulate the expression of various clock genes [148]. In the reciprocal direction, glucocorticoids released by the HPA axis phase shift the circadian rhythm of clock-related genes in peripheral clocks and the tissues/organs they act on [149]. Over-stimulation of the HPA stress system leads to abnormally elevated glucocorticoid levels, which stimulate gluconeogenesis, glycogenolysis, degradation of proteins into amino acids, and eventually, insulin resistance and obesity [150]. Given that the HPA axis and central clock system work in tandem, dysfunction of the clock system affects the rhythmic secretion of glucocorticoids, thus contributing to metabolic abnormalities [151]. The clock and stress systems are crucial to survival, and bidirectional dysregulation of these systems may have pathological consequences [152]. The glucocorticoid receptor (GR) is a clock-controlled gene, which underscores another important association between the circadian system and prostate cancer. 

AAs are exposed to greater stress across their lifetime due to factors such as racial discrimination, socioeconomic status, and stressful life events [153,154,155,156,157]. Compelling evidence suggests that glucocorticoid receptor (GR) signaling is activated by anti-androgen therapy and plays a role in the progression of mCRPC [127]. This mechanistic link has relevance in tumor resistance to anti-androgen therapy and other treatments of mCRPC [32]. In AA men, who are at higher risk for developing mCRPC, greater cortisol levels are positively associated with exposure to chronic stressful events [33]. Using PCa cell lines from racially diverse prostate tumors, it was found that glucocorticoids upregulated two stress oncoproteins related to PCa therapeutic resistance, Clusterin (CLU) and lens epithelium-derived growth factor p75 (LEDGF/p75), while glucocorticoid inhibition blocked this effect. Additionally, AA PCa tissues exhibited greater GR transcript expression compared to European American tissues [33]. Further research is needed to determine the relative contributions of racial disparities (such as the greater exposure to stress for AA men) along with genetic factors to GR transcript expression. Taken together, increased exposure to stressful life events and the difference in GR transcript expression suggest that glucocorticoid signaling may play a contributing role in poor PCa outcomes for the AA population. The glucocorticoid receptor (GR) is a growing target of anti-androgen therapy resistance in mCRPC patients, given that increased GR signaling is one mechanism through which PCa cells evade AR blockades and apoptosis [158]. While glucocorticoids are typically administered in combination with chemotherapy and other treatments because of their role in decreasing toxic side effects and suppressing adrenal androgen production, upregulated GR signaling can promote PCa cell growth in the absence of androgens [159]. Targeting GR expression may improve PCa treatment outcomes. Selective GR modulators inhibited GR transcriptional activity and decreased GR-mediated tumor cell viability post-AR blockade [131]. Lastly, an antagonist of serum and glucocorticoid-regulated kinase 1 (SGK1), an androgen-regulated target gene, was found to block AR-mediated LNCaP cell growth [130], suggesting that the inhibition of GR-related pathways may have a therapeutic benefit in PCa [131,160]. Future research is required to determine if GR-specific antagonists that do not affect the AR could prevent enzalutamide resistance in CRPC progression. 

Obesity-related Biological Pathways. CRD has been shown to exacerbate metabolic disease and obesity, which negatively impact PCa outcomes [161]. Several hypotheses for the link between obesity and CRD have emerged. CRD is associated with the upregulation of hormones, including orexin, growth hormone (increased levels in the daytime), and cortisol (increased levels at night), which contribute to insulin resistance and glucose intolerance [162]. CRD tends to increase ghrelin, a factor that promotes food intake, while decreasing leptin, which induces satiety [163]. From a behavioral perspective, inadequate sleep provides an additional window of time for consuming food at night and promotes daytime lethargy and less physical activity, increasing the risk of obesity [164]. Animal studies suggest that the improper timing of food intake, as well as high-fat diets (HFD), disrupt leptin levels, leading to overeating and sleep deprivation, and thus play a role in the development of obesity, cancer, and metabolic consequences [51]. High-fat diets (HFDs) are known to dampen the circadian rhythms in clock genes, in which the energy consumed during the rest phase is heightened [164]. Recent studies have shown that a high-fat diet can alter the period length of clock genes and reduce the amplitude of metabolic gene expression in the liver, fat, and hypothalamus [52]. The relationship between CRD and altered metabolisms is bidirectional. Shift workers who experienced circadian rhythm misalignment developed insulin resistance, inverted cortisol rhythms, and increased blood pressure [165]. Thus, CRD may exacerbate metabolic disease over time. Mutations in core clock genes can also have significant effects on the risk for metabolic disease: *Clock* mutant mice are obese and develop hyperglycemia, while mice with mutations in the *Period* and *Bmal1* genes display increased adiposity [166]. Several metabolic and hormonal consequences of obesity, including excess levels of adipokines such as leptin and IL-6, insulin resistance, inflammation, and elevated IGF-I levels, are implicated in CRD [167]. 

The coupled effects of circadian disruption on these obesity-related pathways may exacerbate the risk for PCa. Leptin and adiponectin are the most common adipokines functionally contributing to the migration and proliferation of epithelial cells. CRD is associated with reduced levels of leptin during times of wakefulness and leptin resistance [168,169,170]. Mechanistic evidence points to leptin-enhanced PCa cell migration, while inhibiting the MAPK and PI3K signaling blocked migration, suggesting that MAPK and PI3K are potential pathways through which leptin affects PCa progression [34]. In addition to adipokines, circadian gene variants also impact cytokine and insulin growth factor (IGF) levels. Several circadian gene variants, including *NPAS2, PER1, CSNK1E, PER3, and CRY2*, have altered IGF-1 expression. A study of 241 men elderly men revealed that the PER3 variant was associated with higher serum levels of IGF1, supporting a role for circadian gene variants in hormone-related cancers, including PCa [133]. 

Melatonin Inhibition. Light-induced CRD leads to the suppression of melatonin or phase shifting in the timing of melatonin onsetting and offsetting, which has been shown to contribute to greater risk for prostate cancer [171]. Melatonin has antitumor properties, including targeting inflammation and the energy metabolism. Functionally, melatonin’s action is mediated by two membrane-bound G-protein-coupled receptors, MT1 and MT2, which are expressed in human prostate cells [35]. Studies in in vitro and in vivo preclinical models demonstrated the antitumor activity of melatonin against PCa. In LNCaP human PCa cells, melatonin blocked nuclear translocation of the AR. Furthermore, in the TRAMP mice model, melatonin increased survival rates by 33% when given at the beginning of the tumor progression stage [172]. Core clock components, namely CLOCK, PER2, and BMAL1, are disrupted in PCa. However, melatonin treatment restored the expression of circadian genes in Pca cells and resulted in an increase in *PER2* and *CLOCK* and a decrease in *BMAL1*. PER2 overexpression in Pca cells resulted in decreased cell growth and apoptosis, suggesting that decreased PER2 levels in PCa tissue may promote tumor progression [125]. Melatonin inhibits proliferation in LNCaP and VCaP PCa cells in an AR-dependent manner. Specifically, melatonin led to the downregulation of AR signaling and upregulation of p27 [135]. Moreover, melatonin limits glycolysis through facilitative glucose transporters (GLUT/SLC2A) as well as the pentose phosphate pathway in androgen-sensitive and -resistant tumors, suggesting the reduction in glucose uptake as a mechanism via which melatonin impairs PCa growth [137]. These findings suggest that circadian regulators, cortisol, and melatonin serve as potential biomarkers for detecting advanced stage PCa [136]. Further investigations are required to determine whether melatonin treatment results in the greater efficacy of androgen-depletion therapies. 

## 2. Conclusions and Future Directions

Circadian rhythms—cycles in the body that occur in approximately 24 h periods—are involved in many aspects of daily functioning, including the sleep–wake cycle, digestion, hormonal activity, immune function, and body temperature. Mounting evidence shows that CRDs contribute to prostate cancer risk, along with other adverse health outcomes, including obesity, immune dysfunction, and stress-related disorders. The AA population may have a heightened vulnerability to developing CRDs due to environmental conditions (such as living in neighborhoods characterized by high nocturnal noise pollution and greater exposure to ALAN), the prevalence of nightshift work, and structural disadvantages (including socioeconomic stress, workplace discrimination, housing segregation, as well as limited access to nutritious food and quality education). Chronic conditions such as obesity and stress-related disorders are exacerbated by CRDs and may also increase the risk for prostate cancer for the AA population. We review the literature on the interplay among race/ethnicity, CRD, and prostate cancer, highlighting complex interactions that may account for racial disparities and investigating the contribution of circadian-related pathways to prostate cancer outcomes. The racial disparities in prostate cancer may be mediated by CRDs, which have been shown to contribute to PCa progression through circadian gene variants, stress- and obesity-related biological pathways, and melatonin inhibition. 

Epidemiological studies of night shift workers provide strong evidence for the association between prostate cancer and gene variants, including *BMAL1*, *NPAS2*, *CRY2* and *PER2*. Population-based genetic association studies of mostly White and Chinese groups have found that alterations in at least one SNP in each of the core circadian genes was correlated with PCa risk. Racial disparities in both the risk and progression of PCa, as well as in CRD risk, suggest that further research must focus on the AA population and other vulnerable groups. In addition, CRD affects the rhythmic secretion of glucocorticoids released by the HPA system, which has negative implications for mCRPC, especially for AA men. The GR plays a complex role in PCa: in the presence of active AR signaling, glucocorticoids have antitumor effects; however, androgen deprivation increases GR expression, which allows GR to transcriptionally express AR target genes and GR target genes, contributing to tumor resistance to anti-androgen therapy. Therefore, understanding the effects of GR across different cell types will help inform the best strategies to reduce the pro-tumorigenic effects of GR and the optimal use of GR antagonists. Moving forward, the downstream targets of the GR signaling pathway which are responsible for resistance should be determined. Future clinical studies should examine the effectiveness of anti-androgen therapy in combination with GR antagonists—including the dosage, side effects, and specificity of GR-modulators—in overcoming GR-driven resistance to enzalutamide treatment. The clinical benefits of GR-modulators in prostate cancer treatment hold great significance for improving PCa outcomes for AA men. Additionally, CRD may also exacerbate obesity, which contributes to poor PCa outcomes. Increased levels of leptin due to CRD are associated with greater PCa cancer cell migration, potentially contributing to epithelial/mesenchymal transition, metastasis, or angiogenesis. The different subtypes of PCa that may be sensitive to obesity-related changes must be classified and studied. Future research is needed to validate the ability of leptin to stimulate the proliferation of androgen-resistant PCa cells and serve as a novel biomarker to assess PCa aggressiveness. Lastly, melatonin, a key regulator of circadian rhythms, has also been shown to have an inhibitory role in PCa due to its antitumor properties, ability to restore levels of key clock components that are disrupted in PCa, and its role in blocking the nuclear translocation of AR. Melatonin may be a novel therapeutic target for the treatment of PCa. Future studies are necessary to elucidate the exact molecular mechanisms of MT1 and MT2 in PCa cells. To further exploit the therapeutic benefits of melatonin, the assessment of long-term outcomes from the clinical use of melatonin after combined hormone-radiation treatment for PCa patients of different risk groups is an essential and urgent task. The clock–cancer connection in prostate cancer and circadian-related therapies provide a potential novel opportunity to mitigate racial disparities in prostate cancer risk, progression, and treatment outcomes through the optimization of current treatment modalities and reduction in therapeutic resistance. 

## Figures and Tables

**Figure 1 cancers-14-05116-f001:**
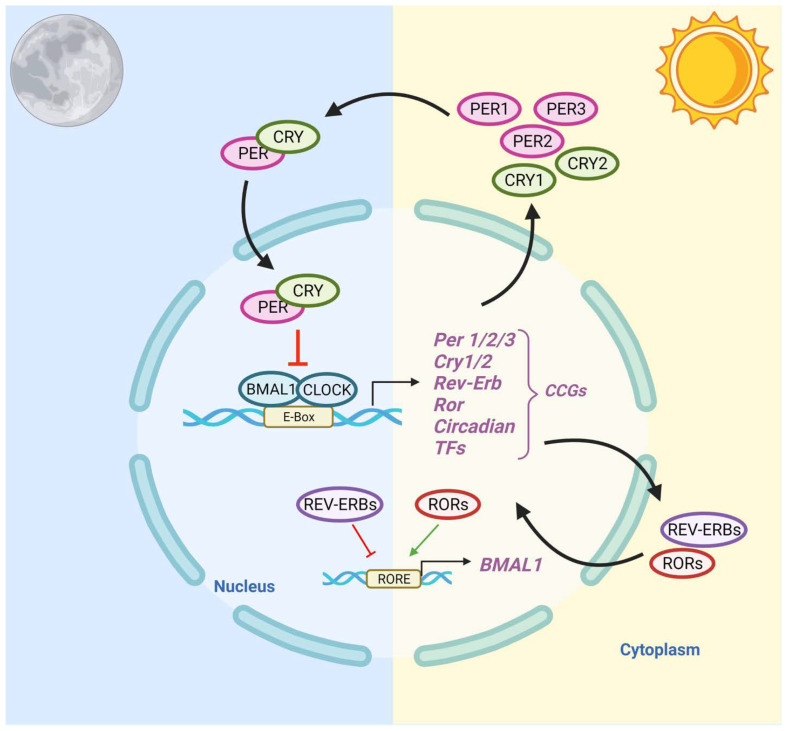
Schematic Representation of the Circadian Clock Transcriptional–Translational Feedback Loops. The main feedback loop is comprised of activator proteins Brain and Muscle ARNT-Like 1 (BMAL1) and Circadian Locomotor Output Cycles Kaput (CLOCK), and two repressor proteins, Period (PER) and Cryptochrome (CRY). BMAL1 and CLOCK heterodimerize and bind to the E-box, which activates the transcription of CRY (1-2), PER (1-3), RORα, REV-ERBα, and Clock transcription factors (TFs). The primary negative feedback occurs when CRY and PER accumulate and dimerize in the cytoplasm and translocate to the nucleus to inhibit the BMAL1: CLOCK. In the secondary feedback loop, RORα and REV-ERBα activate and inhibit the transcription of BMAL1.

**Figure 2 cancers-14-05116-f002:**
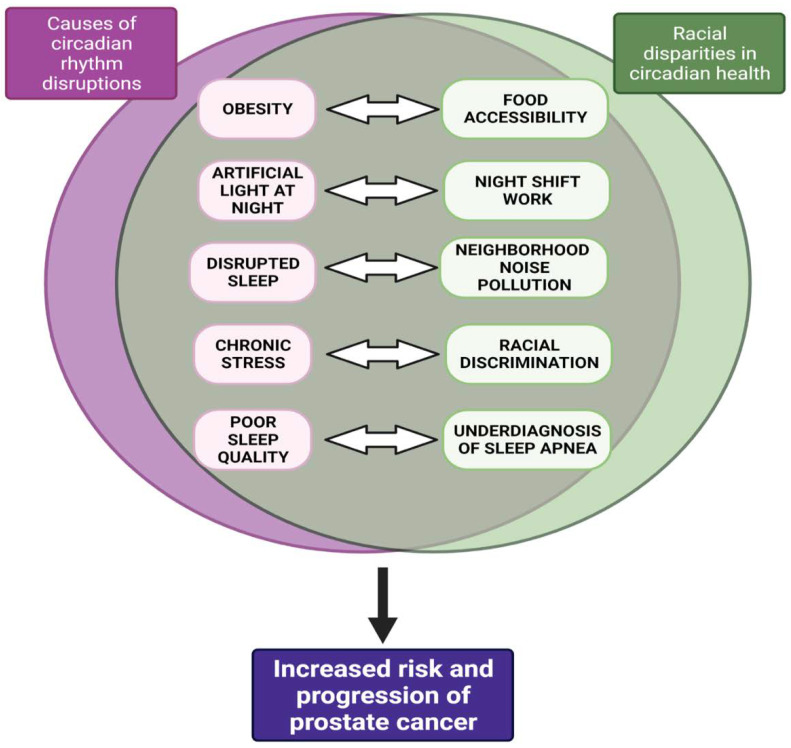
Overlapping Risk Factors for Circadian Rhythm Disruptions and Racial Disparities in Circadian health regarding Prostate Cancer Risk. The right-hand side of the Venn diagram describes environmental, structural, and health-related factors that contribute to the racial disparities in circadian health. Specifically, these factors disproportionately affect the AA population compared to other racial/ethnic groups. The left-hand side outlines the causes of circadian rhythm disruptions. The arrows inside the Venn diagram indicate that there is a bidirectional relationship between the causes of CRDs and social determinants of health. Taken together, the top half of the model illustrates how the African American population is disproportionately at risk for developing CRDs. The bottom half of the model focuses on the downstream effects of circadian rhythm disruption on prostate cancer, both in incidence and aggressiveness.

**Table 1 cancers-14-05116-t001:** Biological Pathways Linking CRDs and Prostate Cancer. The table provides a summary of pathways in which CRDs contribute to prostate cancer incidence and outcomes. The upward arrow refers to pathways, genes, or proteins that are upregulated, while the downward arrow refers to downregulated pathways, genes, or proteins.

CRD-related Pathways	Effect on Prostate Cancer	Therapeutic Targets	
**Circadian Gene Variants**	Per 1, Per 2, and Clock ↓ and Bmal1 ↑ [124]	Increased risk of Pca[124]	Melatonin ↑ Per 2 and Clock and ↓ Bmal1 levels [125]	
Per 1-3, CSNK1E,Cry 1-2, BMAL1,CLOCK andNPAS2 SNPs [29]	Greater risk of aggressive Pca [29]	Overexpression of Per 1 and Per 2 induces growth inhibition [124]	
Per 3 pathway [31]	Regulation of PCSCs[31]	CRY1 levels promote DNA repair and cancer survival [30]	
Per1 decreased AR-related genes in the presence of DHT [126]	
**Stress**	Glucocorticoids ↑ CLU andLEDG/p75 [127]	Pca therapy resistance[127]	RU-486 and cyproterone acetate revert docetaxel resistance [128]	
↑ GR transcript expression following anti- androgen therapy[129]	Tumor progression in mCRPC [129]	SGK1 antagonist blocks AR-mediated growth [130]	
SGRMs ↓ GR transcriptional activity and ↓ GR-mediated tumor cell viability post-AR blockade [131]	
GR upregulation[132]	Bypass AR blockade[132]	
**Obesity**	↑ Leptin [34]	Migration [34]	MAPK and PI3K inhibits migration of Pca cells in the presence of leptin [34]	
NPAS2, Per 1, Per3, Cry 2, andCSNK1E [133]	Altered IGF-1 and androgen [133]	
**Melatonin Inhibition**	↓ 6-STM serum levels [134]	Increase risk for advanced Pca [134]	Melatonin inhibits Pca cell proliferation, ↓ AR signaling and ↓ p27 pathway [135]	
↓ melatonin: cortisol ratio and PSA levels [136]	Increased risk for primary and advanced Pca [136]	Melatonin inhibits glycolysis and the pentose phosphate pathway [137]

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
