# Peer review of "Circadian Rhythm Disruption as a Contributor to Racial Disparities in Prostate Cancer"

_cancers, 2022, doi:10.3390/cancers14205116_

Round 1

Reviewer 1 Report

Dasari et al.,

Lines 92-93: Light during the day signals the SCN to increase body temperature, heart rate and blood pressure, and suppress melatonin production. In the evening, melatonin production increases, and the opposite effect occurs

This statement is over generalized and too simplified, light does not send signals and the SCN does not directly control all the stated physiological changes

Lines 96-97: The primary TTFL is made up of positive feedback mechanism and negative feedback mechanism

The transcriptional-translational feedback loop is only a negative feedback mechanism; indeed, it contains both positive and negative regulators, but the loop cannot be both positive and negative feedback.

Paragraph 104-121:

Referencing times of day as “the morning” and “at night” can cause confusion to the reader since the mechanisms are in opposing times of the day in models such as mice versus humans, best would be to indicate the “active phase” or the “inactive phase.” Alternatively, the authors should indicate that they are describing human profiles.

Line 203: ALAN has been found to be associated with PCa risk.

Citation is missing.

Lines 209-210: Exposure to light at night affects melatonin levels

Citation is missing.

Table 1: Circadian gene variants- WNT/beta-Catenin pathway

This pathway should not be included among circadian gene variants as these are not circadian genes, it may be best to label this reference as “Per3”

Lines 379-383: ARNTL, a circadian rhythm core component

ARNTL is synonymous to BMAL1 (Arntl = gene name, BMAL1 = protein name). As used in these lines, ARNTL is depicted as if it were a unique protein, authors should clarify that these are the same transcription factor.

Paragraph 399-427:

It would be relevant to note that GR is a clock-controlled gene and thus another association between the clock and prostate cancer.

Lines 428-429: CRD has been shown to exacerbate metabolic disease and obesity, which negatively impact PCa outcomes.

Citation is missing.

Author Response

Point 1: Lines 92-93: Light during the day signals the SCN to increase body temperature, heart rate and blood pressure, and suppress melatonin production. In the evening, melatonin production increases, and the opposite effect occurs

This statement is over generalized and too simplified, light does not send signals and the SCN does not directly control all the stated physiological changes

Response 1: This is a valid point and in response, we have specified that these physiological changes occur in diurnal species. We have also replaced “light during the day sends signals” with “light phase” and “in the evening” with the “dark phase”. Finally, we have removed the direct reference to the SCN.

Point 2: Lines 96-97: The primary TTFL is made up of positive feedback mechanism and negative feedback mechanism

The transcriptional-translational feedback loop is only a negative feedback mechanism; indeed, it contains both positive and negative regulators, but the loop cannot be both positive and negative feedback.

Response 2: This point has been noted and corrected in the revised manuscript. We have described the TTFL as a self-regulating mechanism that is comprised of a positive arm and a negative arm, in accordance with the reviewer’s suggestion.  

Point 3: Paragraph 104-121:

Referencing times of day as “the morning” and “at night” can cause confusion to the reader since the mechanisms are in opposing times of the day in models such as mice versus humans, best would be to indicate the “active phase” or the “inactive phase.” Alternatively, the authors should indicate that they are describing human profiles.

 Response 3: We agree with the reviewer and apologize for the ambiguity of this description. We have now indicated that we are describing human profiles. We have indicated that the “light phase” refers to morning, and “dark phase” refers to the evening in the human circadian system.

Point 4: Line 203: ALAN has been found to be associated with PCa risk.

Citation is missing.

Response 4: The citation has been added.

Point 5: Lines 209-210: Exposure to light at night affects melatonin levels

Citation is missing.

Response 5: The citation has been added.

Point 6: Circadian gene variants- WNT/beta-Catenin pathway

This pathway should not be included among circadian gene variants as these are not circadian genes, it may be best to label this reference as “Per3”

Response 6: We agree with the reviewer on this point and have correct the table accordingly.

Point 7: Lines 379-383: ARNTL, a circadian rhythm core component

ARNTL is synonymous to BMAL1 (Arntl = gene name, BMAL1 = protein name). As used in these lines, ARNTL is depicted as if it were a unique protein, authors should clarify that these are the same transcription factor.

Response 7: We have replaced “ARNTL” with “BMAL1 (also known as Arntl)” for clarity, in accordance with reviewer’s suggestion.

Point 8: Paragraph 399-427:

It would be relevant to note that GR is a clock-controlled gene and thus another association between the clock and prostate cancer.

Response 8: A sentence has been at the end of the first paragraph in the Stress-related Biological Pathways section to include the information from this comment.

Point 9: Lines 428-429: CRD has been shown to exacerbate metabolic disease and obesity, which negatively impact PCa outcomes.

Citation is missing.

Response 9: The citation has been added.

Reviewer 2 Report

General Comments

The authors present a review paper that argues for a significant contribution of Circadian Rhythm Disruption (CRD) to the known increased burden of prostate cancer for African American men.  Their review is based on the assertion that the current scientific body of evidence implicates CRD as a driver of prostate cancer risk and progression.  They also argue that AA men are particularly vulnerable to CRDs due to greater exposure to night shift work, artificial light at night, noise pollution, racial discrimination, and socioeconomic disadvantages. While their review helps frame this working hypothesis, the underlying premise of a CRD – prostate cancer association and AA men being more likely vulnerable to CRD is tenuous at best.

The last comprehensive review (Wendeu-Foyet & Menegaux CEBP 2017) of the epidemiologic evidence for an association between CRD and prostate cancer concluded, “Evidence of a possible association between night shift work and prostate cancer remains to date inconclusive“. The authors should provide a more balanced appraisal of scientific evidence supporting a CRD-prostate cancer association.

Most of the evidence for AA being at greater risk for CRD in this review that the authors present is indirect – can the authors cite publications that directly show a racial difference in CRD risk?

Can the authors discuss the results of one or more original research reports that link obesity and/or stress with CRD?

Can the information in table 1 be incorporated into an overall conceptual model figure that supports the underlying assertion of this review that AA are at greater risk for CRD and this contributes in part to the observed disparate prostate cancer outcomes?

Specific Comments

Lines 53-58; 70-71; 328-329; 399-400: None of the declarative statements in these sections are supported by referenced publications.

Statement in lines 169-170 seems to contradict earlier statement in lines 166-167 concerning pilots and prostate cancer mortality.

Author Response

Point 1: The last comprehensive review (Wendeu-Foyet & Menegaux CEBP 2017) of the epidemiologic evidence for an association between CRD and prostate cancer concluded, “Evidence of a possible association between night shift work and prostate cancer remains to date inconclusive“. The authors should provide a more balanced appraisal of scientific evidence supporting a CRD-prostate cancer association.

Response 1: We acknowledge the limitations within the current literature on CRD and prostate cancer risk. We have integrated the existing evidence to the best of our knowledge. In relation to the reviewer’s comment on night shift work and prostate cancer, in section 1.2.1 Causes of CRDs, we state that “despite evidence implicating a positive correlation between night shift work and PCa, some reports have found no such association,” and include citations of studies that have not found an association.   

Point 2: Most of the evidence for AA being at greater risk for CRD in this review that the authors present is indirect – can the authors cite publications that directly show a racial difference in CRD risk?

Response 2: We take a socio-ecological approach to consider the complex interplay between various social determinants of health, the causes and consequences of circadian rhythms and its potential association with the racial disparities in prostate cancer. We integrate the existing evidence in the literature to the best of our knowledge. We aim to provide a framework that elucidates the overlapping risk factors and the complex connection between racial disparities in both circadian health and prostate cancer. Among many publications cited in the manuscript, Eastman, C.I et al, Lunsford-Avery et al, provide evidence for a racial difference in CRD risk.

Point 3: Can the authors discuss the results of one or more original research reports that link obesity and/or stress with CRD?

Response 3: In the manuscript, we discuss Shi, S.Q. et al’s report on the effect of circadian disruption on insulin resistance and obesity. We discuss Nader, N. et al’s report on the connection between the HPA stress axis and the circadian system. 

Point 4: Can the information in table 1 be incorporated into an overall conceptual model figure that supports the underlying assertion of this review that AA are at greater risk for CRD and this contributes in part to the observed disparate prostate cancer outcomes?

Response 4: We believe that the information presented in Table 1 provides a comprehensive account (based on evidence/published studies) supporting mechanisms underlying the potentially higher risk for CRD in African American patients and contributing to the observed disparate prostate cancer outcomes. It is critical for the full appreciation of the concept and scale of the review, that the information is presented on a Table rather than a Figure, as the evidence and “full picture” is still evolving.

Point 5: Lines 53-58; 70-71; 328-329; 399-400: None of the declarative statements in these sections are supported by referenced publications.

Response 5: In lines 53-58, referenced citations have been included, which provide support for the connection between CRD and racial disparities in circadian health. In lines 70-71, the relevant publications (which appear in section 1.4 The Consequence of CRD on Prostate Cancer Risk and Progression) have been added. In lines 328-329, the source for that sentence can be found directly after the second sentence after the semicolon (Wendeu-Foyet, M.G et al. Circadian genes polymorphisms, night work and prostate cancer risk: Findings from the EPICAP study). In lines 399-400, the relevant sources have been cited that provide evidence that African Americans and other marginalized groups experience greater psychosocial stressors across their lifetime.

Point 6: Statement in lines 169-170 seems to contradict earlier statement in lines 166-167 concerning pilots and prostate cancer mortality.

Response 6: We appreciate the reviewer identifying the contradictory information that detracts from the flow of scientific evidence. We have revised line 169-170 in accordance to the source cited, Raslau et al, which asserts that pilots were at greater 

Round 2

Reviewer 2 Report

Overall, the manuscript is improved and the authors were generally responsive to the reviewer comments. In response to my request for a discussion of results from one or more original research reports that link obesity and/or stress with CRD the authors state that  they mention Shi, S.Q. et al’s report on the effect of circadian disruption on insulin resistance and obesity and Nader, N. et al’s report on the connection between the HPA stress axis and the circadian system.  However, neither of these reports are discussed in the Obesity-related Biological Pathways section.